# Pheochromocytomas and Paragangliomas: New Developments with Regard to Classification, Genetics, and Cell of Origin

**DOI:** 10.3390/cancers11081070

**Published:** 2019-07-29

**Authors:** Karen Koopman, Jose Gaal, Ronald R. de Krijger

**Affiliations:** 1Martini Hospital, 9728 NT Groningen, The Netherlands; 2Department of Pathology, Isala Hospital, 8025AB Zwolle, The Netherlands; 3Department of Pathology, University Medical Center Utrecht, 3584 CX Utrecht, The Netherlands; 4Princess Maxima Center for Pediatric Oncology, 3584 CS Utrecht, The Netherlands

**Keywords:** pheochromocytoma, paraganglioma, tumor classification, scoring system, SDHB, PASS, GAPP

## Abstract

Pheochromocytomas (PCC) and paragangliomas (PGL) are rare neuroendocrine tumors that arise in the adrenal medulla and in extra-adrenal locations, such as the head, neck, thorax, abdomen, and pelvis. Classification of these tumors into those with or without metastatic potential on the basis of gross or microscopic features is challenging. Recent insights and scoring systems have attempted to develop solutions for this, as described in the latest World Health Organization (WHO) edition on endocrine tumor pathology. PCC and PGL are amongst the tumors most frequently accompanied by germline mutations. More than 20 genes are responsible for a hereditary background in up to 40% of these tumors; somatic mutations in the same and several additional genes form the basis for another 30%. However, this does not allow for a complete understanding of the pathogenesis or targeted treatment of PCC and PGL, for which surgery is the primary treatment and for which metastasis is associated with poor outcome. This review describes recent insights into the cell of origin of these tumors, the latest developments with regard to the genetic background, and the current status of tumor classification including proposed scoring systems.

## 1. Introduction

Pheochromocytomas (PCC) are rare neuroendocrine tumors that, by definition, arise in the adrenal medulla; however, the term paraganglioma (PGL) is reserved for tumors that arise from sympathetic ganglia in the thorax, abdomen, and pelvis as well as from parasympathetic paraganglia in the head and neck area. Although these tumors are rare, with an incidence of approximately five per million, they are relevant from various perspectives, including pathophysiology and genetic background. Such tumors also have a relationship with various other tumors, such as neuroblastomas, that also originate from the adrenal medulla and extra-adrenal sympathetic tissues as well as tumors co-occurring in various hereditary tumor syndromes, for instance, medullary thyroid carcinoma in MEN2, renal cell carcinoma in VHL disease, and gastrointestinal stromal tumors (GIST) in the context of succinate dehydrogenase subunit gene (collectively called *SDHx*) mutations.

PCC and PGL are composed of chromaffin cells, deriving from the adrenal medulla and sympathetic ganglia [1]. Previously, these chromaffin cells were thought to develop from a common neural crest precursor cell that migrated to the dorsal aortic area and also produced sympathetic neurons [2,3]. However, several recent studies have challenged the notion of the common ancestry of chromaffin cells and sympathetic neurons. These studies have shown that the progenitor cells have partly overlapping but distinct expression profiles [4,5,6,7]. Chromaffin cells, which form the chief component of the adrenal medulla, have been shown to derive from peripheral glial stem cells, known as Schwann cell precursors. These have a limited capability for expansion and migrate along preganglionic nerves to reach their adrenal medulla destination. These new findings are supported by genetic ablation experiments and further supported by distinct gene expression programs, such as detection by single cell RNA sequencing experiments. However, it has been shown recently that divergence is not absolute and that Schwann cell precursors appear to be responsible for a small but relevant fraction of sympathetic neurons; additionally, a proportion of chromaffin cells may derive from migrating neural crest cells [8]. It will be challenging to relate these findings to the development of PCC and PGL as well as to relate this work to the origin of neuroblastomas, the infantile and pediatric tumor originating from the same locations.

In this review, two major issues that have dominated the PCC and PGL field with regard to patient management and research will be discussed, with an emphasis on the latest developments. One is the genetic background of PCC and PGL, which have been shown to be among the tumors with the highest rate of hereditary predisposition. Currently, up to 35–40% of PCC and PGL carry germline mutations, divided amongst more than 20 different genes but functionally clustered into two major clusters [9]. We will focus on the more recently discovered genes and on SDHx genes, which are relevant for their association with metastatic potential. The other issue concerns the histopathological classification, in which the distinction between benign (non-metastatic) and potentially metastatic tumors on the basis of one or multiple criteria with sufficient sensitivity and specificity to be clinically useful has been virtually impossible. Currently, as discussed in the most recent version of the WHO volume on Endocrine Tumors, all PCC and PGL are considered to have some metastatic potential [10]. The current need is to recognize those tumors that have a sufficiently high metastatic potential to warrant close follow-up or treatment. Two recent histopathological scoring systems, correlating with metastatic potential, hold some promise for this purpose, including the Pheochromocytoma of the Adrenal gland Scaled Score (PASS) and the Grading system for Pheochromocytoma and Paraganglioma (GAPP) [11,12], both of which will be discussed.

## 2. Genetics

Knowledge of molecular abnormalities of PCC and PGL has rapidly grown over the past decades. Germline and somatic mutations in at least 20 susceptibility genes have been reported in hereditary and sporadic tumors. Currently, up to 40% of PCC/PGL have a hereditary origin and can be identified in another 30% somatic mutations. It has been shown that germline as well as somatic mutations are largely mutually exclusive; however, several examples of concurring mutations have been reported [9,10,13,14].

On the basis of their transcriptional profile, PCC and PGL have been divided into two clusters. Cluster 1 is characterized by the pseudohypoxia pathway and includes, among others, PCC/PGL with *VHL* and *SDHx* mutations. Cluster 2 is characterized by activation of kinase signaling pathways and includes, among others, PCC/PGL with *RET* and *NF1* gene mutations [15,16,17,18,19,20,21]. With the identification of new somatic drivers, including gene fusions, a new cluster 3 has been recognized, representing tumors with Wnt signaling activation. This cluster includes PCC/PGL with *MAML3* fusions, most commonly *UBTF-MAML3*, and somatic mutations in *CSDE1* and may be associated with more aggressive disease [17,22,23,24]. A fourth cluster with tumors with adrenocortical features has also been suggested, but this cluster is less well defined and lacks consistent mutations [14,22]. In addition, somatic mutations in more common cancer-related genes, such as *TP53* and *BRAF*, have been associated with PCC/PGL [10,15]. Furthermore, “disease modifying genes” have been recognized, including somatic mutations in *ATRX*, which mostly co-exist with mutations in *SDHB* or *IDH1/2*, with a suggested synergistic effect on tumorigenesis and tumor progression [15,24,25,26]. A discussion of all the gene mutations, proposed mechanisms, and associated conditions is beyond the scope of this review. We will focus on the driver mutations in Clusters 1 and 2, discuss the most common members with special attention to the *SDHx* family, and highlight some recently identified gene mutations, as seen in Table 1. 

### 2.1. Cluster 1

This cluster represents germline or somatic gene mutations resulting in a dysfunctional hypoxic (pseudo-hypoxic) response with a central role for hypoxia inducible factor (HIF)1 alpha and HIF2 alpha, which are the main components of the response to low oxygen levels. Under normal conditions, HIF1 alpha and HIF2 alpha are in an inactivated state. Under hypoxic conditions or in the context of a dysfunctional system, HIFs are stabilized and activate target genes inducing angiogenesis, metabolism, apoptosis, and proliferation. The cluster can be divided into two subgroups. The *VHL/EPAS1*-related subgroup includes mutations in genes directly involved in activation of the HIF signaling pathway such as *VHL germline (g), somatic (s), EPAS1/HIF2 alpha (g,s)* and *EGLN1/2 (g)*. The TCA-cycle-related subgroup includes mutations in genes encoding energy metabolism enzymes such as *SDHx (g), FH (g), MDH2 (g)* and *IDH1/2 (s).* Inactivating mutations lead to an accumulation of TCA metabolites causing epigenetic changes in HIF stability and are associated with a hypermethylated phenotype [13,16,17,20,21,23,24,25,27]. 

#### 2.1.1. VHL/EPAS1-Related

The *VHL* gene encodes two proteins, pVHL30 and pVHL19. pVHL plays a critical role in degradation of HIF alpha, and pVHL dysfunction results in a pseudohypoxic response. The *VHL* gene is known to cause von Hippel-Lindau (VHL) disease, an autosomal dominant familial tumor syndrome characterized by multiple benign and malignant tumors including retinal and central nervous system hemangioblastomas, renal cysts, clear cell renal cell carcinomas (RCC), pancreatic cysts, pancreatic neuroendocrine tumors, endolymphatic sac tumors, and epididymal cystadenomas. VHL disease shows a genotype–phenotype correlation; VHL type 2 is characterized by missense mutations and is linked to PCC, which is observed in 10–26% of VHL patients. Germline mutations in VHL account for 5–10% of hereditary PCC/PGL, and somatic *VHL* mutations have been found in about 10% of sporadic PCC/PGL [10,28,29].

The *EPAS1/HIF2 alpha* gene encodes for HIF2 alpha and mutations lead to its reduced degradation and stabilization. Somatic *EPAS1/HIF2 alpha* mutations have been found in 5–10% of PCC/PGL. In addition, germline mutations have been identified, but they comprise <1% of PCC/PGL. *EPAS1/HIF2 alpha* mutations are also linked to polycythemia and somatostinoma [30,31,32]. 

The *EGLN1* and *EGLN2* genes encode for PHD2 and PHD1, respectively. PHD proteins are involved in the degradation of HIFs by hydroxylation. Germline mutations in *EGLN1* and *EGLN2* have been identified in <1% of PCC/PGL and are also associated with polycythemia [33,34]. 

#### 2.1.2. Tricarboxylic Acid TCA Cycle-Related

Mutations in the *SDHx* family are currently the most common germline mutations found in hereditary PCC/PGL and account for about 20–30% of hereditary cases. Somatic mutations have rarely been reported [35]. *SDHx*-related PCC/PGL are caused by mutations in five genes of the *SDHx* family, *SDHA*, *SDHB*, *SDHC*, *SDHD,* and *SDHAF2*, which encode the corresponding proteins. The SDH complex, composed of the four subunits SDHA–D, is a key component of the TCA cycle and the electron transport chain and is located at the inner mitochondrial membrane. SDHAF2 is essential in the assembly of the complex. The complex catalyzes the oxidation of succinate to fumarate. Loss of SDH function results in accumulation of succinate, which inhibits the prolyl hydroxylation of HIF1 alpha, resulting in a hypoxic response [36,37]. Mutations in the *SDHx* family are predominantly linked to extra-adrenal PGLs. *SDHB* mutations are associated with abdominal and thoracic PGL and show a 30–70% risk of metastasis. *SDHD* and *SDHA* mutations are mostly found in abdominal, thoracic, head, and neck PGLs. *SDHC* and *SDHAF2* mutations are associated with head and neck PGLs [10]. Mutations in the *SDHD* and *SDHAF2* genes, both located on chromosome 11, show a parent-of-origin-dependent inheritance effect in which mutations almost exclusively cause disease after paternal transmission. Somatic loss of maternal chromosome 11 plays an important role in this effect according to the Hensen model [38,39]. Mutations in the *SDHx* family have also been associated with other tumors such as SDH-deficient GIST, SDH-deficient RCC, and SDH-deficient pituitary adenoma [10]. 

If the SDH complex becomes instable by dysfunction of any of the components, the SDHB subunit is degraded in the cytoplasm. The loss of SDHB expression can be detected by SDHB protein immunohistochemistry (IHC); this can be used to detect tumors with mutations in one of the five genes. In addition, loss of SDHA IHC can detect tumors with *SDHA* germline mutations [40,41,42,43]. The use of SDHB/SDHA IHC has been suggested as a supplementary tool in guiding molecular genetic testing. Correct interpretation of SDHB/SDHA IHC is important because false positive as well as false negative results can lead to incorrect interpretations or inappropriate genetic testing. SDHB IHC positivity comprises granular cytoplasmatic staining with the same intensity as the internal positive control cells (non-tumoral cells, e.g., lymphocytes, endothelial cells, fibroblasts). If the tumor cells show absent, less intense, or blush-like non-granular staining, SDHB IHC should be regarded as negative, as seen in Figure 1. Furthermore, interpretation of SDHB IHC in tumor areas with clear cytoplasm is not recommended because staining might appear negative. SDHB IHC negativity has also been shown in tumors with *SDHC* promoter methylation in the absence of *SDHx* germline mutations. In addition, SDHB negative PCC/PGL without known pathogenic *SDHx* mutations or *SDHC* promoter methylation have been described, suggesting that SDHB IHC might be of additional value in the assessment of *SDHx* genetic variants of unknown significance [42,43].

#### 2.1.3. FH, MDH2, IDH1/IDH2

The *FH* gene encodes fumarate hydratase, a TCA enzyme which converts fumarate into malate. Mutant FH results in accumulation of fumarate leading to PHD inactivation and HIF stabilization. Mutations in this gene have classically been associated with hereditary cutaneous and uterine leiomyomatosis and a specific variant of RCC, which is regarded as a separate entity by the WHO [44]. Recently, germline mutations have been identified in PCC/PGL, accounting for <5% of PCC/PGL [45]. FH IHC can show loss of expression in *FH*-mutated PCC/PGL [46]. In addition, low tumor levels of 5-hmC (resembling those in SDHB-deficient tumors) and positive 2SC staining have been detected in tumors with *FH* mutations, suggestive of altered DNA methylation and protein succination, respectively [47]. The *MDH*2 gene encodes MDH2, another TCA enzyme that catalyzes the reversible oxidation of malate to oxaloacetate. MDH2 dysfunction leads to accumulation of metabolites and a hypermethylator phenotype. Germline mutations have been identified in less than 1% PCC/PGL and show undetectable 5-hmC staining. The isodehydrogenase (IDH) enzymes IDH1 and IDH2, encoded by *IDH1* and *IDH2*, convert isocitrate to alpha-ketoglutarate in the TCA cycle. Mutations result in the production of 2-hydroxyglutarate instead of alpha-ketoglutarate, and its accumulation activates the hypoxic pathway. Somatic *IDH1* and *IDH2* mutations are typically present in low-grade and secondary high-grade gliomas and have rarely been found in PGLs [48,49]. 

#### 2.1.4. Recently Identified PCC/PGL Genes

Germline mutations in other genes with a TCA-related function and/or a hypermethylated phenotype have been recently described in 1% or fewer of PCC/PGL and might be included in this cluster. The *SLC25A11* gene encodes the mitochondrial 2-oxoglutarate/malate carrier and germline mutations have been identified in abdominal as well as head and neck PGLs and may be associated with metastases [50]. The *IDH3B* gene encodes IDHB3, an enzyme involved in the oxidation of isocitrate to alpha-ketoglutarate in the TCA cycle, and a germline mutation was identified in a jugular PGL [48]. GOT2 is another mitochondrial enzyme involved in amino acid metabolism as well as the urea and TCA cycle; additionally, a germline mutation in the encoding *GOT2* gene has been found in metastatic abdominal and thoracic PGL with a high succinate/fumarate ratio in tumor cells [48]. The *DNMT3A* gene plays a role in DNA methylation during embryonic development and germline mutations have been found in multifocal PGLs [51]. The *DLST* gene encodes a mitochondrial protein that belongs to the 2-oxoacid dehydrogenase family. Germline mutations have recently been found in PCC/PGL with an expression and methylation profile suggestive of the pseudohypoxic pathway. In addition, positive DLST immunostaining was detected in tumors with TCA-cycle or *EPAS1* mutations [52].

### 2.2. Cluster 2 

The common denominator for Cluster 2 is the activation of kinase signaling pathways. It encompasses germline or somatic mutations in *RET (g/s)*, *NF1 (g/s), MAX (g/s)*, *MEN1 (g)*, *TMEM127 (g)*, *H-RAS (s)*, and *KIF1B (g/s)*. These mutations are associated with the activation of the RAS-RAF-ERK, PI3K-AKT-mTOR (*RET, HRAS, NF1, TMEM127, KIF1B*), and MYC-MAX (*MAX*) kinase signaling pathways. The activation or deregulation of these pathways leads to uncontrolled proliferation, growth, and cell survival [13,17,21,23,24,25]. 

The *RET* proto-oncogene encodes a transmembrane tyrosine kinase. When ligand binds to the RET receptor or if there is an activating mutation, a signaling cascade is triggered through the PI3 kinase pathway to regulate cell proliferation and apoptosis. Activating germline mutations in the *RET* gene cause the multiple endocrine neoplasia type 2 (MEN2) syndrome characterized by the development of PCC and medullary thyroid carcinoma. PCC occur in 40–50% of patients with MEN2 and the risk of PCC is associated with specific *RET* mutations. Clues to MEN2-related PCC include the presence of multiple, bilateral tumor nodules and adrenal medullary hyperplasia. Germline *RET* mutations have been detected in 5% of hereditary PCC/PGL and somatic mutations in about 5% of sporadic tumors [29,53,54].

The *NF1* gene encodes neurofibromin 1, which is a negative regulator of the RAS intracellular signaling pathway. Inactivating mutations in the *NF1* gene disrupt this inhibitory effect. *NF1* is the most common somatically mutated gene in PCC/PGL, accounting for 20–40% of sporadic PCC/PGL. However, only approximately 6% of patients with the autosomal dominant NF1 syndrome, characterized by germline *NF1* mutations, develop PCC, and germline mutations account for <5% of all PCC/PGL. Interestingly, up to 16% of NF1-related PCC show a composite histological phenotype of typical PCC intermixed with areas of ganglioneuroblastoma or ganglioneuroma [9,55]. 

*TMEM 127* is a tumor suppressor gene linked with the mTOR kinase pathway, and germline mutations have been found in less than 5% of PCC/PGL. It encodes a transmembrane protein that functions as a negative regulator of mTOR. Mutant forms of the protein are nonfunctional and cause cytoplasmic localization of the protein. Notably, mutant alleles show relatively low penetrance [56,57].

*MAX* is a tumor suppressor gene with a regulatory role in the MYC-MAX-MD1 network. Truncated *MAX* mutations can cause dysregulation of the mTOR pathway. Germline mutations, which may exhibit a parent-of-origin effect with paternal transmission, as well as somatic mutations have been found in PCC/PGL, both in less than 5% of cases [58,59]. Although MAX IHC has been described in the literature, with the potential to detect *MAX* mutations in patients in case of absent immunostaining, comparable to negative SDHB staining, this has not found widespread application [59].

*H-RAS* is a proto-oncogene that encodes H-RAS protein, which can bind to GTP and activate the RAS/RAF/ERK signaling pathway leading to cell proliferation. Missense somatic mutations have been found in 5–10% of sporadic PCC/PGL [60,61].

The *KIF1B* gene encodes for two proteins isoforms, KIF1B alpha and KIF1B beta. KIF1B alpha is involved in the transport of mitochondria, and KIF1B beta in the transport of synaptic vesicle precursors. In addition to its transport function, KIF1B beta plays a role in apoptosis and dysfunctional KIF1B beta due to mutations in the *KIF1B* gene and can lead to tumorigenesis. The *KIF1B* gene might be one of the most frequently somatically mutated genes in PCC/PGL, with somatic mutations seen in up to 20% of cases, and also germline mutations have been described in fewer than 1%. *KIF1B* mutant PCC/PGL shows a similar transcription profile as *RET* and *NF1* mutant PCC/PGL [62,63]. 

The *MEN1* gene codes for the menin protein, which interacts with transcription factor JunD. JunD is a functional component of the AP1 transcription factor complex. It has been proposed to protect cells from p53-dependent senescence and apoptosis. Germline mutations in the *MEN1* gene lead to MEN1 syndrome, characterized by various types of endocrine and non-endocrine tumors, including pituitary adenomas, hyperplastic parathyroids, and neuroendocrine pancreatic tumors, with a high penetrance of disease but PCC/PGL are rare and account for <1% of PCC/PGL [64,65,66]. 

## 3. Histopathological Classification

All PCC/PGL are believed to exhibit some metastatic potential, but only a subset of these tumors will actually metastasize. Until 2004, malignancy was defined by either metastasis or on the basis of extensive local invasion. The 2004 WHO classification defined malignant PCC by the development of metastasis. Due to incongruity, confusion was noted and the current WHO classification encourages the use of the terms “metastatic” and “non-metastatic”. Many histomorphological features and immunohistochemical markers have been studied to find biomarkers to differentiate between metastasizing and non-metastasizing PCC/PGL. However, no single histomorphological or immunohistochemical feature has indicated metastatic potential. Therefore, several algorithms, on the basis of multiple histologic properties, have been described to detect potential for aggressive behavior. 

In 1990, Linnoila examined 120 PCC/PGL and developed a statistical model to predict malignancy. According to this model, there was a 95% probability that more than 70% of tumors could be classified correctly on the basis of four factors: extra-adrenal location, coarse nodularity, confluent tumor necrosis, and absence of hyaline globules. The majority of malignant PCC/PGL had two or three of these features (71%) while 89% of benign tumors had none or one [67]. Unfortunately, malignancy in this study was defined by regional or distant metastases and/or extensive local invasion.

### 3.1. Pheochromocytoma of the Adrenal Gland Scaled Score (PASS)

Thompson described the Pheochromocytoma of the Adrenal gland Scaled Score (PASS), which consists of twelve parameters: large nests or diffuse growth, central or confluent necrosis, high cellularity, cellular monotony, tumor cell spindling, >3 mitotic figures/10 high power fields, atypical mitotic figures, extension into adipose tissue, vascular invasion, capsular invasion, profound nuclear pleomorphism, and nuclear hyperchromasia. A maximum score of 20 points is obtained when all features are present. Tumors with a score <4 were considered to have no metastatic potential. Tumors with a score ≥4 were considered to have an increased metastatic potential, as seen in Table 2 [11].

Commonly agreed upon features are the presence of atypical mitosis, necrosis, capsular involvement, lymphovascular invasion, and extension into adipose tissue. However, significant interobserver and intraobserver variability is described in scoring the presence or absence of high cellularity, nuclear pleomorphism, and hyperchromasia [68]. A meta-analysis of the PASS algorithm found a sensitivity of 97% and a specificity of 68%. The positive predictive value (PPV) was 31%, and the negative predictive value (NPV) 99%. These findings suggest that a PASS score of <4 is highly indicative of a benign clinical course, but for tumors with PASS score of ≥4, the predictive value with regard to disease course is limited [69]. 

### 3.2. Grading of Adrenal Pheochromocytoma and Paraganglioma (GAPP)

Although PASS only applies to PCC, the Grading of Adrenal Pheochromocytoma and Paraganglioma (GAPP) is a tool for PCC as well as PGL. It is based on histological pattern, cellularity, comedo-type necrosis, capsular invasion, vascular invasion, Ki67 labelling index and catecholamine type. On the basis of a maximum score of 10 points, tumors are graded as well differentiated (0–2 points), moderately differentiated (3–6 points), and poorly differentiated (7–10 points), as seen in Table 2. The five-year survival rates in these groups are 100% for well differentiated, 66.8% for moderately differentiated and 22.4% for poorly differentiated tumors. Kimura also found that Ki67 was significantly different between the metastatic and non-metastatic group [12]. Using a 3% cutoff, Ki-67 shows a sensitivity, specificity, positive predictive value, and negative predictive value of 55.6%, 94.9%, 83.3%, and 82.2%, respectively [70].

A sensitivity of 50% and specificity of 80% has been described for PCC with GAPP. In addition, the PPV was 5% and NPV was 99%. For PGL, the sensitivity was 100% and the specificity was 68%, with a PPV of 29% and a NPV of 100% [69]. 

Both PASS and GAPP do not account for the effect of underlying somatic or germline mutations, whether these are “low risk driver mutations” or “high risk driver mutations”. For example, PCC in MEN 2A typically have a low risk of metastases, with an estimated prevalence of 3% [71]. However, in almost 50% of the tumors, PASS scores of ≥4, which are indicative of potential aggressive behavior, were found. In addition the same tumors also displayed an elevated GAPP and were scored as moderately differentiated. The most frequently reported histopathological characteristics in these tumors were the presence of large nests, diffuse growth, and high cellularity. Thirty percent of the MEN 2A-related PCC displayed a Ki67 labelling index of >3%, making it the most common GAPP criterion. These findings suggest that both algorithms are inadequate to determine metastatic potential in MEN2A-related PCC [72].

Recurrence and metastases are strongly associated with *SDHB* mutations; the presence of *SDHB* mutations should be considered a risk factor for metastases or recurrence [73]. As described above, SDHB IHC is a strong indication for tumors with *SDHx* mutations. Loss of SDHB staining is not seen either in prognostically unfavorable *SDHB*-mutated tumors or in all tumors with *SDHx* mutations; however, SDHB IHC negativity by itself has also been correlated with metastases [42,43]. In a GAPP follow-up study, SDHB IHC served as a surrogate marker for genetic testing. All tumors lacking SDHB staining were intermediate or high grade, and 10 out of 13 had metastasized [12]. Additionally, *MAML3* fusion gene variants, leading to Wnt signaling upregulation, as seen in Cluster 3, described above, are associated with metastases. In the same study, Ki67 was found to be correlated with metastatic disease. Interestingly, the tumor with the highest Ki67 expression was *MAML3* fusion-positive [22]. Another study assessed promoter methylation density of tumor suppressor genes with frequent hypermethylation in cancer. Fourteen PGL, six metastatic and eight non-metastatic, as well as two distant metastases were significantly methylated at one or more of the following gene promoters, *RASSF1A, NORE1A, p16INK4A, RARB, DCR2, CDH1*, and *APC*. Methylation of three or more promoters was found in five PGL, of which four were metastatic [74]. Despite the promising histomorphological, immunohistochemical, and molecular biomarkers described above, there are no criteria with sufficient sensitivity and specificity to accurately predict metastatic potential.

## 4. Discussion/Conclusions

Recently, PCC and PGL have proven to be enigmatic tumors with regard to the prediction of their clinical behavior; they are also exciting tumors with regard to their pathophysiology and genetic background. Major progress has been made in various fields, although the survival of metastatic PCC and PGL is still bleak. In the field of PCC and PGL genetics, the number of genes involved continues to increase, with a particular focus on enzymes and other proteins that are involved in the TCA cycle. With the advent of whole exome and whole genome sequencing, more such genes may be detected, all of which may potentially converge along the same pathophysiological pathway. Whole exome sequencing is expected to become the standard of genetic analysis in the next 5–10 years, allowing rapid detection of mutations in PCC and PGL patients and families. IHC, for instance, with SDHA and SDHB antibodies, may serve as a confirmatory technique with regard to the classification of variants of unknown significance. With regard to classification, the most important new developments can be found in the 2017 edition of the WHO volume on Endocrine Tumors, in which all PCC and PGL are now considered to have some metastatic potential and that the pathological grading system for PCC and PGL (GAPP), developed by Kimura, has been independently validated. In addition, a modified GAPP (M-GAPP) has been proposed, incorporating loss of SDHB staining as a criterion, which seems logical in light of the increased metastatic risk related to *SDHB* mutations [75]. However, the loss of SDHB staining occurs with all *SDHx* mutations, and increased risk has not been noted for *SDHA, SDHC,* and *SDHD* mutations. Both PASS and GAPP have a low PPV but a high NPV, suggesting that these models are good at ruling out but poor at predicting metatastatic potential.

Finally, although outside the scope of this review, the improved knowledge of the genetic background and the genes involved in the pathogenesis of PCC and PGL will allow for the development of targeted therapies for those tumors that cannot be resected surgically or that have metastasized, potentially improving prognosis for these patients.

## Figures and Tables

**Figure 1 cancers-11-01070-f001:**
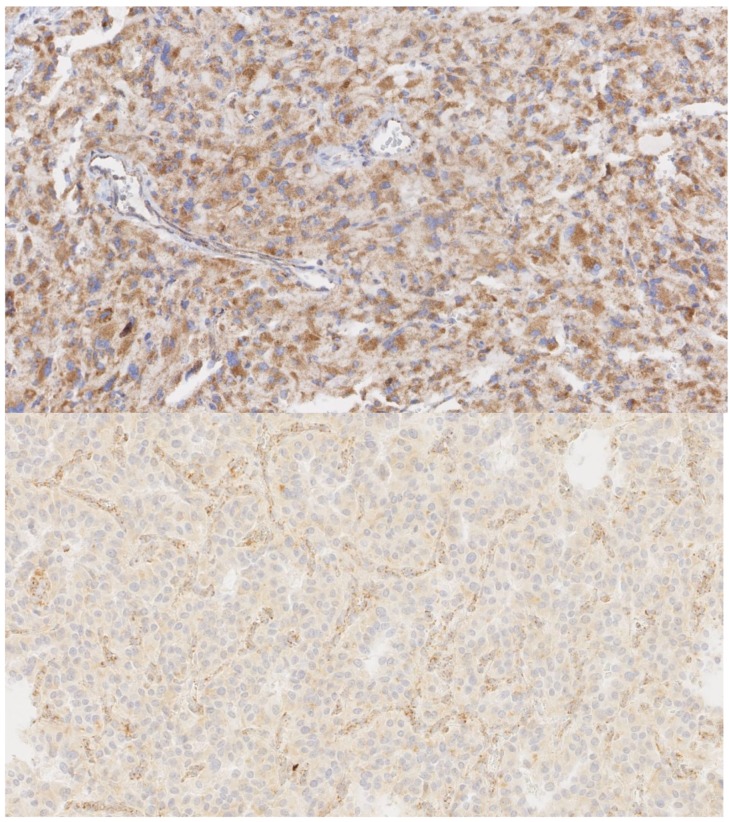
SDHB immunohistochemistry of paraganglioma with normal staining pattern in upper panel, showing strong diffuse granular staining in the cytoplasm of all cells. Lower panel shows absent staining in tumor cells. Note the remaining cytoplasmic granular staining in pre-existent normal endothelial cells and fibroblasts. Magnification: 400×.

**Table 1 cancers-11-01070-t001:** Clinicopathologic and genetic characteristics of PCC/PGL-related genes.

GENE	Germline %	Somatic %	Tumor Type	Metastatic Risk %	Syndrome/Other Tumors
**Cluster 1-VHL/EPAS1**					
*VHL*	5–10	10	PCC>>PGL	5	VHL syndrome
*EPAS1/HIF2 alpha*	<1	5–10	PCC/ATPGL	29	Polycythemia, somatostatinoma
*EGLN1/EGLN2*	<1	-	PCC/ATPGL	?	Polycythemia
**Cluster 1-TCA**					
*SDHx*	20–30	<1			
*SDHA*	<5		PGL	Low	GIST, RCC, PA
*SDHB*	5–10		ATPGL>>HNPGL/PCC	30-70	GIST, RCC, PA
*SDHC*	<5		HNPGL	Low	GIST, RCC
*SDHD*	5–10		PGL>PCC	<5	GIST, RCC, PA
*SDHAF2*	<1		HNPGL	Low	
*FH*	<5	-	PCC/PGL	>50	Leiomyoma, RCC
*MDH2*	<1	-	ATPGL	?	
*IDH1/IDH2*	-	<1	PGL	?	Low grade glioma
Recently identified genes					
*SLC25A11*	<1	-	PGL	High?	
*IDH3B*	<1	-	HNPGL	?	AML
*GOT2*	<1	-	ATPGL	High?	
*DNMT3A*	<1	-	HNPGL	?	AML
*DLST*	<1	-	PCC/PGL	?	
**Cluster 2**					
*RET*	5	5	PCC	<5	MEN2 syndrome
*NF1*	<5	20–40	PCC	12	NF1 syndrome
*TMEM127*	<5	-	PCC>PGL	Low	RCC
*MAX*	<5	<5	PCC/PGL	10	Renal oncocytoma
*H-RAS*	-	5–10	PCC	Low	
*KIF1B*	<1	20	PCC	?	Neuroblastoma
*MEN1*	<1	-	PCC/HNPGL	?	MEN1 syndrome

AML: acute myeloid leukemia; ATPGL: abdominal or thoracic paraganglioma; GIST: gastrointestinal stromal tumor; HNPGL: head and neck paraganglioma; MEN1: multiple endocrine neoplasia type 1; MEN2: MEN type 2; NF1 syndrome: neurofibromatosis type 1; PA: pituitary adenoma; PCC: pheochromocytoma; PGL: paraganglioma; RCC: renal cell carcinoma; VHL syndrome: von Hippel-Lindau syndrome.

**Table 2 cancers-11-01070-t002:** Histopathological classification systems.

PASS		GAPP	
Pheochromocytoma	yes	Pheochromocytoma	yes
Paraganglioma	no	Paranganglioma	yes
**Parameters**	**Score**	**Parameters**	**Score**
Nuclear hyperchromasia	1	**Histological Pattern**	
		Zellballen	0
		Large and irregular cell nest	1
		Pseudorosette (even focal)	1
Profound nuclear pleomorphism	1	**Cellularity**	
		Low (less than 150 cells/U *)	0
		Moderate (150–250 cells/U *)	1
		High (more than 250 cells/U *)	2
Capsular invasion	1	**Comedo Necrosis**	
		Absence	0
		Presence	2
Vascular invasion	1	**Vascular or Capsular Invasion**	
		Absence	0
		Presence	1
Extension into periadrenal adipose tissue	2	**Ki67 Labelling Index**	
		<1%	0
		1–3%	1
		>3%	2
Atypical mitotic figures	2	**Catecholamine Type**	
		Epinephrine type (E **, or E + NE ***)	0
		Norepinephrine type (NE, or NE + D ****)	1
		Non-functioning type	1
>3 mitotic figures/10 high-power field	2	**Total**	10
Tumour cell spindling	2		
Cellular monotony	2		
High cellularity	2		
Central or confluent tumour necrosis	2		
Large nests or diffuse growth (>10% of tumour volume)	2		
**Total**	20		

A PASS score <4 or ≥ 4 suggest non-metastatic potential versus metastatic potential respectively. Using GAPP tumors are graded as well differentiated (0-2 points), moderately differentiated (3-6 points) and poorly differentiated (7-10 points). * U: numer of tumour cells in a square of 10 mm micrometer observed under high power magnification (×400). ** E: epinephrine. *** Norepinephrine. **** D: Dopamine.

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
