# Peer review of "Pheochromocytomas and Paragangliomas: New Developments with Regard to Classification, Genetics, and Cell of Origin"

_cancers, 2019, doi:10.3390/cancers11081070_

Round 1
Reviewer 1 Report
This paper is a brief review /update concerning several aspects of pheochromocytoma paraganglioma. The information is useful and is mostly current. I offer only a few minor comments.
1. While the title mentions cell of origin, this area is given relatively short shrift. I suggest some clarification and elaboration. On page 1 line 42 “extra-adrenal sympathetic lineages” should be changed to “sympathetic neurons” because the Furlan paper cited did not deal with extra-adrenal paraganglia. In addition, a followup paper that does deal with extra-adrenal paraganglia (Kastriti ME, Kameneva P, Kamenev D, Dyachuk V, Furlan A, Hampl M, Memic F, Marklund U, Lallemend F, Hadjab S, et al. 2019 Schwann Cell Precursors Generate the Majority of Chromaffin Cells in Zuckerkandl Organ and Some Sympathetic Neurons in Paraganglia. Front Mol Neurosci 12 6) backs off a bit from the sharp neuron/chromaffin cell divergence and allows that small contributions of migratory neural crest cells do contribute to chromaffin cell populations and that Schwann cell precursors (SCPs) contribute to a “small but significant” proportion of sympathetic neurons. It should also be mentioned that the SCPs themselves are derived from the neural crest, by way of the dorsal root ganglia. Chromaffin cells therefore are still derived from the neural crest, athough by a very indirect route. It has been suggested that SCPs might constitute a pool of neural crest-like cells employed for expansion and diversification of neural crest-derived tissues after the neural crest itself ceases to exist (Furlan A, Adameyko I. Schwann cell precursor: a neural crest cell in disguise? Dev Biol. 2018;444 Suppl 1:S25-S35. doi: 10.1016/j.ydbio.2018.02.008. PubMed PMID: 29454705).
2. Page 2 line 55 Please change “malignant potential” to “metastatic potential” . Same for other lines where malignant is used. It should be noted that WHO 2017 encourages this terminology to avoid previous ambiguity caused by a competing definition based on local invasion, and that this ambiguity is present in impotant early studies (e.g. Linnoila et al, ref 61).
Author Response
Reviewer 1: We thank the reviewer for the excellent suggestion to elaborate some more on the cell or origin of PCC and PGL and we have adapted the paragraph of the introduction (lines 40-56) accordingly, and inserted a series of references as well.
Reviewer 1: We acknowledge that metastatic potential is the preferred term to be used and have adapted this throughout the text.
Reviewer 2 Report
This is a very nice overview of the current knowledge of the genetic background of pheochromocytoma and paragangliomas. The histopathological scoring systems add additional prediction about aggressiveness and malignant potential. Most clinicians will be familiar with the PASS scoring system, but not so much with the GAPP system.
Overall, a well written and comprehensive review which will be helpful for clinicians and scientists alike.
Author Response
We thank this reviewer for the comments. We added a table with the GAPP and PASS criteria.
Reviewer 3 Report
The authors provide an overview of the classification, genetics and cell of origin of pheochromocytomas and paragangliomas.
This is a review paper, and many of the review in the text can be elaborated using figures, tables and diagrams. E.g. the cells of origin, clusters, signaling pathways, comparing and contrasting syndromic gene causes. It only has one table which is insufficient.
There are numerous stylistic and grammatic errors in this manuscript throughout, which significantly detracts from its readability and overall message.
Author Response
We thank the reviewer for the constructive comments and have specified below how we have dealt with these comments:
We agree with the reviewer that we have been sparing on figures and tables. While there are many review papers that have already presented a plethora of such tables and figures, we have now expanded the current table and added a table with an overview of the PASS and GAPP criteria and added a figure of SDHB immunohistochemistry.
We regret that this reviewer had a suboptimal reading experience while reviewing our manuscript. We have had the manuscript corrected by a native speaker and have made numerous small changes throughout the entire text, assuming that the current text meets the standards of the reviewer and the journal.
Round 2
Reviewer 3 Report
Manuscript is improved. All genes should be italicized.
Author Response
We wouuld like to thank this reviewer.
All genes are now in Italic.